# Inflammasome-independent NLRP3 function enforces ATM activity in response to genotoxic stress

Mélanie Bodnar-Wachtel[1,2,3,4,\*], Anne-Laure Huber[1,2,3,4,\*], Julie Gorry[1,2,3,4], Sabine Hacot[1,2,3,4], Delphine Burlet[1,2,3,4], Laetitia Gérossier[1,2,3,4], Baptiste Guey[1,2,3,4], Nadège Goutagny[1,2,3,4], Birke Bartosch[1,2,3,4], Elise Ballot[5], Julie Lecuelle[5], Caroline Truntzer[5], François Ghiringhelli[5], Bénédicte F Py[6], Yohann Couté[7], Annabelle Ballesta[8], Sylvie Lantuejoul[4,9], Janet Hall[1,2,3,4], Agnès Tissier[1,2,3,4], Virginie Petrilli[1,2,3,4]

NLRP3 is a pattern recognition receptor with a well-documented role in inducing inflammasome assembly in response to cellular stress. Deregulation of its activity leads to many inflammatory disorders including gouty arthritis, Alzheimer disease, and cancer. Whereas its role in the context of cancer has been mostly explored in the immune compartment, whether NLRP3 exerts functions unrelated to immunity in cancer development remains unexplored. Here, we demonstrate that NLRP3 interacts with the ATM kinase to control the activation of the DNA damage response, independently of its inflammasome activity. NLRP3 down-regulation in both broncho- and mammary human epithelial cells significantly impairs ATM pathway activation, leading to lower p53 activation, and provides cells with the ability to resist apoptosis induced by acute genotoxic stress. Interestingly, NLRP3 expression is down-regulated in non-small cell lung cancers and breast cancers, and its expression positively correlates with patient overall survival. Our findings identify a novel non-immune function for NLRP3 in maintaining genome integrity and strengthen the concept of a functional link between innate immunity and DNA damage sensing pathways to maintain cell integrity.

## Introduction

NLRP3 belongs to the NOD-like receptor (NLR) family of cytosolic pattern recognition receptors involved in innate immunity (Martinon & Tschopp, 2005). Upon sensing pathogen-associated molecular patterns or damage-associated molecular patterns, such as nigericin or ATP, respectively, NLRP3 triggers the assembly of the inflammasome, a multi-protein complex composed of the adaptor apoptosis-associated speck-like containing a caspase activation and recruitment domain (ASC) and the effector caspase-1 (Mariathasan et al, 2006; Pétrilli et al, 2007a; Schroder & Tschopp, 2010). Activated caspase-1 then induces the maturation of the pro-inflammatory cytokines IL-1β and IL-18, and eventually, upon gasdermin-D cleavage, a proinflammatory cell death called pyroptosis (Mariathasan & Monack, 2007; Shi et al, 2015b; Broz & Dixit, 2016). The NLRP3 inflammasome is mostly expressed in myeloid cells where its biological functions have been well characterized. Deregulation of NLRP3 activity by mutations or exogenous stimulation provokes many inflammatory disorders such as cryopyrinopathies, gouty arthritis, or Alzheimer disease (Mangan et al, 2018). In the context of cancer, its role appears complex as it was reported to either promote or inhibit tumor growth (Karki et al, 2017; Petrilli, 2017). Whether NLRP3 exerts functions unrelated to immunity in cancer remains unexplored.

Incorrectly or unrepaired DNA double-strand breaks (DSBs) are sources of genomic instability, a factor known to promote tumorigenesis (Negrini et al, 2010; Abbas et al, 2013). Toxic DNA DSBs arise from both exogenous sources, for instance ionizing radiation (IR) exposure, and endogenous sources, such as DNA replication stress. One of the key proteins that orchestrates the rapid cellular response to DSBs is the ataxia-telangiectasia–mutated (ATM) kinase. In resting cells, ATM is present as an inactive dimer. Once recruited to DSBs via the action of the MRE11-RAD50-NBS1 (MRN) complex, ATM autophosphorylates, monomerizes, and initiates a vast cascade of post-translational modifications that are essential for the DNA damage response (DDR) (Bakkenist & Kastan, 2003). For instance, phosphorylation of the histone variant H2AX on Ser139 (γH2AX) is one of the earliest events in the DDR and is crucial for an efficient recruitment of DNA repair proteins to DNA strand breaks (Paull et al, 2000; Burma et al, 2001; Celeste et al, 2002; Xie et al,

[1]INSERM U1052, Centre de Recherche en Cancérologie de Lyon, Lyon, France  [2]CNRS UMR5286, Centre de Recherche en Cancérologie de Lyon, Lyon, France  [3]Université de Lyon, Université Claude Bernard Lyon 1, Lyon, France  [4]Département de Biopathologie, Centre Léon Bérard, Lyon, France  [5]Département d'oncologie Médicale, INSERM 1231, Université de Bourgogne, Dijon, France  [6]CIRI, Centre International de Recherche en Infectiologie, University Lyon, INSERM, U1111, Université Claude Bernard Lyon 1, CNRS, UMR5308, ENS de Lyon, Lyon, France  [7]Université Grenoble Alpes, CEA, INSERM, UA13 BGE, CNRS, CEA, FR2048, Grenoble, France  [8]INSERM and Université Paris Sud, UMRS 935, Campus CNRS, Villejuif, France & Honorary Position, University of Warwick, Coventry, UK  [9]Département de Pathologie, Pôle de Biologie et de Pathologie, Centre Hospitalier Universitaire, Inserm U823, Institut A Bonniot-Université J Fourier, Grenoble, France

Correspondence: virginie.petrilli@lyon.unicancer.fr
*Mélanie Bodnar-Wachtel and Anne-Laure Huber contributed equally to this work

2004). ATM also phosphorylates many downstream effector proteins including KAP1, p53, and CHK2 that induce effector mechanisms such as the activation of cell cycle checkpoints, apoptosis, or senescence (Smith et al, 2010). The ATM pathway is tightly regulated and any dysregulation in these protection mechanisms facilitates the progression of cells toward malignancy. Indeed, many studies reported that the expression of genes involved in the DDR, including *ATM*, are frequently attenuated during tumor progression, and mutations in these genes confer increased susceptibility to cancer occurrence (Gayther et al, 1995; Bartkova et al, 2005).

Here, we report that NLRP3 expression is down-regulated in human cancers. Importantly, we demonstrate for the first time that NLRP3 interacts with ATM and this interaction is instrumental to reach optimal ATM activation in response to DNA damage. As a consequence of this key regulatory role, NLRP3 loss confers resistance to acute genotoxic stress–induced cell death.

## Results

### NLRP3 is down-regulated in lung and breast cancers (BC)

We were intrigued by genome-wide studies reporting that *NLRP3* is frequently altered in non-small cell lung cancer (NSCLC) and BC (Kan et al, 2010; Hoadley et al, 2013). Interestingly, somatic mutations in *NLRP3* together with 10 other genes including *KEAP1*, *KRAS*, and *STK11* were reported to define a molecular signature associated with a subtype of NSCLC, the lung adenocarcinoma (LUAD) (Hoadley et al, 2013). To further investigate the pattern of NLRP3 expression in NSCLC, we first assembled a panel of 13 NSCLC cell lines and included 3 human immortalized broncho-epithelial cell (HBEC3) lines (Ramirez et al, 2004) for comparison. In most of the cancer cell lines analyzed, including cell lines reported to carry NLRP3 mutations, the NLRP3 protein was barely detectable, whereas ASC or caspase-1 was variably expressed (Fig 1A). Consistently, very low levels of *NLRP3* mRNA were observed by Q-RT-PCR in the NSCLC cell lines in comparison with HBEC3-KT cells (Fig 1B). Interestingly, among the 3 HBEC3 lines (see description in the Materials and Methods section), HBEC3-ET cells did not express NLRP3 and were the only cells to display properties of malignant transformation, as demonstrated by their ability to grow in an anchorage-independent manner (Fig S1A). These results suggest that NLRP3 expression may be down-regulated during tumor progression. To validate these observations, we obtained a set of tumor RNA samples from a cohort of patients with primary NSCLC and control RNAs from adjacent normal lung tissues. As shown in Fig 1C, *NLRP3* mRNA was detectable in normal lung tissues, whereas it was significantly down-regulated in NSCLC tissues. Those observations were confirmed by analyzing TCGA dataset from LUAD and lung squamous cell carcinoma (Fig S1B). Because genome-wide studies reported that the *NLRP3* locus is amplified in BC (Kan et al, 2010), we investigated whether NLRP3 was differentially expressed in human immortalized mammary epithelial cells (HMECs) at different stages of transformation (R. Weinsberg's model Fig S1C) versus cancer cells (Elenbaas et al, 2001; Morel et al, 2008). As for HBEC3 cells, lower levels of NLRP3 protein were observed in the transformed cells (HMLE)

compared with immortalized mammary epithelial cells (HMEC-hTERT), with undetectable levels found in HMLER cells suggesting, as well, a repression of NLRP3 expression during malignant progression (Fig S1C). To address this hypothesis in a second tumor model, the level of NLRP3 protein was assessed in 14 BC cell lines (Fig S1D). Compared with HMEC-hTERT, NLRP3 protein appeared undetectable in 12 of the cell lines with the exception of the HER2 positive (HER2+) JIMT1 and the basal MDA-MB231 cells. Analyses of data from 1,097 breast primary tumors from the TCGA dataset confirmed that NLRP3 expression was lower in the primary tumor compared with normal tissues (132 normal breast tissues) (Fig S1E). Finally, by extending the differential expression analysis to other tissues, we found that in the pancreas, colon, prostate cancers, and more widely pan-cancers, NLRP3 expression was also significantly lower (Fig S1F).

We next investigated whether NLRP3 expression was correlated with patient prognosis. TCGA data analysis showed that high NLRP3 expression in LUAD was positively associated with a better overall survival (OS) and better progression-free interval (PFI) (Fig 1D and E). With respect to BC, elevated NLRP3 expression was positively correlated with patient OS for HER2+ but not basal-like BC subtype (Fig S1G and H), and positively correlated with PFI in HER2+ and basal-like BC (Fig S1I and J). Altogether, these results show a down-regulation of NLRP3 expression in NSCLC and BC, which is correlated with a poorer patient outcome for LUAD, HER2+, and basal BC.

### NLRP3 is required for optimal ATM activation

One hallmark of cancer is genome instability (Hanahan & Weinberg, 2011). Interestingly, we found that low NLRP3 expression positively correlated with genome instability in both LUAD and BC (Fig S2A and B). The ATM pathway is the warrant of genome stability and is frequently inactivated in cancerous lesions (Bartkova et al, 2005; Lantuejoul et al, 2010). Intriguingly, previous works showed that ATM is instrumental to NLRP3 activation in response to infections (Erttmann et al, 2016). We, therefore, questioned whether the reciprocity was true and investigated whether NLRP3 controls ATM activation upon DNA damage. We examined the phosphorylation of ATM substrates in response to etoposide (Eto), a topoisomerase II inhibitor, which is known to induce DNA DSBs, in HBEC3-KT in the presence or absence of NLRP3. KD of NLRP3 led to significant lower phosphorylation (Ser824) levels of KAP1 (P-KAP1) 2–6 h post-treatment with Eto compared with siCTL. The KD of ATM using siATM displayed similar lower P-KAP1 levels after Eto treatment (Figs 2A and S2C). In addition, in siNLRP3 cells, lower levels of P-p53 (Ser15) were observed 2–6 h post-treatment with Eto (Fig 2B). ATM activation by DSBs relies on its monomerization, which coincides with autophosphorylation on Ser1981 (Bakkenist & Kastan, 2003). Using P-ATM foci formation to assess ATM activation, our results showed that NLRP3 down-regulation resulted in fewer P-ATM foci, irrespective of Eto concentrations (Fig S2D and E), and also in fewer γH2AX(Ser139) foci (Fig S2F) consistent with a requirement of NLRP3 for full ATM activation and downstream signaling.

Previous studies reported that ATM dysfunction induces ROS production in cells (Kamsler et al, 2001; Erttmann et al, 2016). In agreement with these observations, NLRP3-depleted cells displayed enhanced ROS content compared with control cells, which did not increase further after the addition of the ATM inhibitor

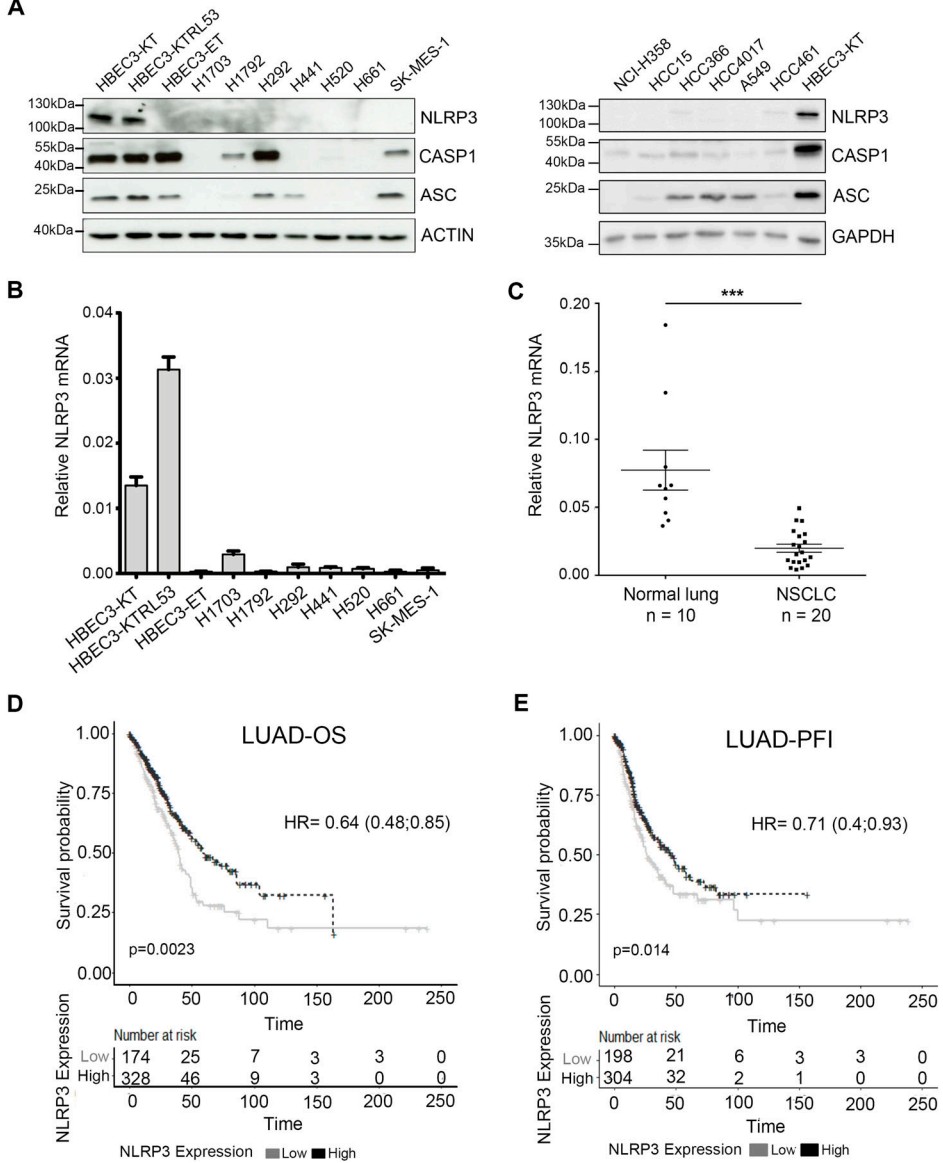

**Figure 1. NLRP3 expression is reduced in human NSCLC compared with healthy tissue.**
**(A)** Protein levels of NLRP3 and the inflammasome components, namely, caspase-1 (CASP1) and ASC, were assessed by immunoblotting in HBEC3 cells and a panel of NSCLC lines. Actin and GADPH served as loading controls. **(B)** Relative *NLRP3* mRNA levels determined by Q-RT-PCR in HBEC3 cells and in a panel of NSCLC cell lines normalized against the ubiquitous esterase-D. Results are representative of more than three experiments. **(C)** Relative *NLRP3* mRNA levels were determined by Q-RT-PCR in a cohort of non-treated primary tumors from NSCLC patients (n = 20) and the corresponding normal lung tissues (n = 10) normalized against esterase-D and *HPRT1*. Data represent mean ± SEM; ***P < 0.001 (*t* test). **(D, E)** Kaplan–Meier plots of patient overall survival (D) and progression free interval (E) in TCGA–LUAD dataset according to NLRP3 expression levels, time is shown in months; patients were stratified according to the cutoff obtained from maximally selected rank statistics.

KU55933 (Fig S2G). Collectively, these results strongly support the notion that NLRP3 is required for optimal ATM activation.

To confirm that NLRP3 is essential for ATM activation upon the direct formation of genotoxic DSBs, we exposed HBEC3-KT cells to IR and assessed the activation of the ATM pathway in the presence or absence of NLRP3. Our results showed that the absence of NLRP3 resulted in lower levels of KAP1 Ser824 and CHK2 Thr68 phosphorylation in response to IR (Fig 2C). In addition, we assessed γH2AX and P-ATM nuclear foci over time. In control cells, the number of nuclear γH2AX foci peaked around 1 h after IR treatment and then decreased with time as the cells underwent DNA repair (Fig 2D). In contrast, in the absence of NLRP3, a significantly lower number of γH2AX foci were initially formed and detected at all subsequent time points. This difference was greatest 1 h post-IR and persisted over the time course of the study. For

P-ATM foci, we observed as early as 15 min post-IR, fewer P-ATM foci in the absence of NLRP3, and the difference remained significant at 1 and 5 h post-IR (Fig 2E). These results support our earlier findings that NLRP3 down-regulation prevents maximal ATM activation after DSB formation. This hypothesis was further investigated using mechanistic mathematical modeling (see the Materials and Methods section). Based on our experimental findings that NLRP3 KD reduced the observed number of P-ATM molecules recruited at DSBs, two theories were assessed: (A) NLRP3 enhances ATM activation, and (B) NLRP3 inhibits ATM deactivation (Fig 2F). Hypothesis A almost perfectly matched the data, whereas hypothesis B did not reproduce the correct dynamics (SSR_A = 0.052, SSR_B = 0.29) (Fig 2G). Model A was able to predict that the ATM activation rate under siNLRP3 conditions $k_{IR}$ was nearly half of that in siCTL cells as the ratio $\frac{k_{IR}^{control}}{k_{IR}^{NLRP3}}$ was equal to 1.66. Thus, the model that best

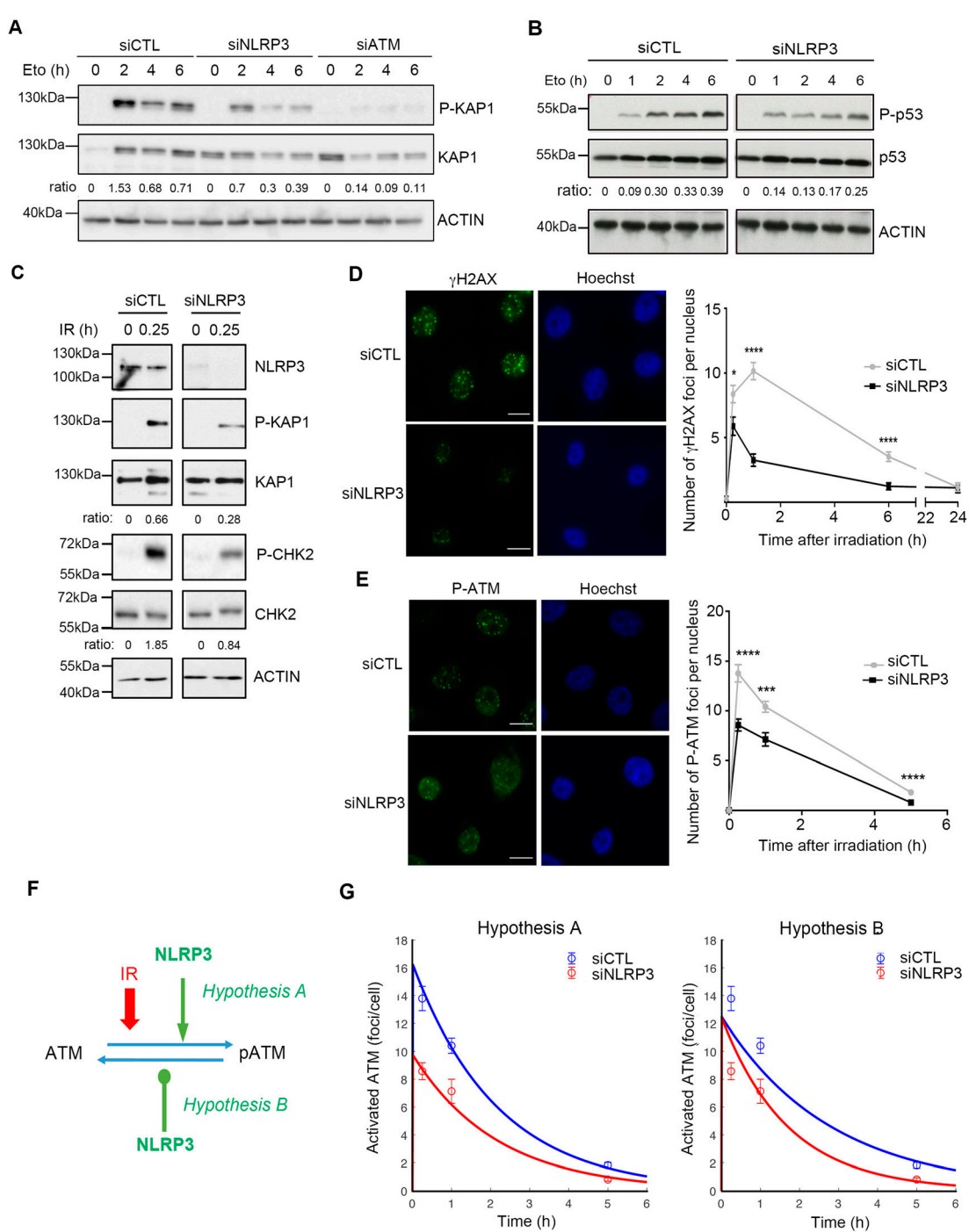

**Figure 2. NLRP3 is instrumental to achieve maximal ATM activation in response to DSBs.**
**(A)** HBEC3-KT transfected with indicated siRNA were treated with 100 µM etoposide (Eto) and collected at different time points, and P-KAP1 (Ser824) was analyzed by immunoblot, the ratio between P-KAP1/KAP1 is shown. One experiment representative of three experiments. **(B)** P-p53 (Ser15) was assessed by immunoblot in HBEC3-KT transfected with siCTL or siNLRP3 treated with 100 µM etoposide. The ratio between P-p53/p53 is shown. One experiment represents three. **(C)** HBEC3-KT transfected with siCTL or siNLRP3 was irradiated (10 Gy), and P-KAP1 and P-CHK2 (Thr68) were analyzed by immunoblot 15 min after IR. The ratio between P-KAP1/KAP1 and P-CHK2/CHK2 is shown. Representative of two experiments. Actin was used as a loading control for all immunoblots. **(D, E)** HBEC3-KT transfected with siCTL or siNLRP3 was irradiated (2 Gy), and γH2AX (D) and P-ATM (E) foci were assessed by IF at the indicated time. Hoechst was used as a nuclear stain. Representative of three independent experiments. Data represent mean ± SEM, ****$P < 0.0001$, ***$P < 0.001$, *$P < 0.05$, (unpaired $t$ test). 64≤ n ≥148, n is the number of nuclei counted. Scale bars 10 µm. **(A, B, F)** Schematic diagram displaying the hypothesis used for mathematical modeling of ATM and NLRP3 interactions, showing the two hypotheses investigated (A) NLRP3 enhances ATM activation and (B) NLRP3 inhibits ATM deactivation. **(G)** Mathematical modeling of ATM and NLRP3 interactions. The curve represents the mathematical model, and the dots represent the experimental data.

recapitulates our data supports the notion that NLRP3 enhances ATM activation.

We then assessed whether NLRP3 was required for ATM activation in the breast cellular model and tested the impact of NLRP3 down-regulation in the mammary epithelial model of HMEC-hTERT cells. As observed in the broncho-epithelial cells HBEC3-KT, NLRP3 depletion in HMEC-hTERT resulted in decreased KAP1 and CHK2 phosphorylation after Eto treatment (Fig S3A). Accordingly, a significantly decreased number of γH2AX foci per nucleus were found in siNLRP3 cells treated with Eto (Fig S3B). In support of our hypothesis, in MDA-MB-231 cells, one of the rare BC cell lines with a sustained NLRP3 expression, NLRP3 depletion using a siNLRP3 approach resulted in decreased levels of P-KAP1 and P-CHK2 compared with the parental cells (Fig S3C).

Finally, to address whether the introduction of NLRP3 into NLRP3-deficient NSCLC tumor cell lines would improve ATM activation, we re-expressed NLRP3 using a doxycycline-inducible system in A549 and H292 cells, and evaluated ATM activation after IR by assessing the number of P-ATM and γH2AX foci. The presence of NLRP3 increased both the ATM activating phosphorylation and γH2AX foci numbers in both cell lines 1 h post-treatment (Figs 3A–D and S4A–E). We also verified whether NLRP3 re-expression in a cellular model commonly used in laboratories, HEK293T cells, which are also deficient in NLRP3, would impact ATM activation. We observed an increase in P-ATM and P-KAP1 upon NLRP3 over-expression and Eto treatment (Fig S4F). Altogether, these results demonstrate that NLRP3 is required to obtain maximal ATM activation upon DSB induction.

## NLRP3 controls the ATM pathway independently of the inflammasome activity

The best-known function of NLRP3 is its ability to form the inflammasome. We thus investigated whether this new role for NLRP3 in DNA damage signaling was inflammasome-dependent. We noticed that we were able to improve ATM activation in A549 cells by re-expressing NLRP3 despite the fact they expressed little caspase-1, suggesting that NLRP3 may act independently of the inflammasome. To test this hypothesis, we KD in HBEC3-KT the inflammasome effector, caspase-1, and assessed γH2AX levels after Eto treatment. No significant difference was observed between siCTL and siCaspase-1 (siCasp1) (Fig 4A). One of the major products of inflammasome activation is the secretion of active IL-1β. To ascertain that DSBs did not induce the NLRP3 inflammasome, we measured the level of IL-1β release upon DSB induction. Monocytic THP1 cells treated with nigericin were used as a positive control of NLRP3 inflammasome activation (Fig 4B). In response to Eto (Fig 4B) or IR (Fig 4C) treatment, no significant level of IL-1β was detected in cell supernatants. Accordingly, no cleaved caspase-1, a hallmark of inflammasome activation, was detected after Eto treatment (Fig 4D). Hence, the presence of DNA DSBs does not activate the catalytic activity of the inflammasome.

## NLRP3-depleted cells are resistant to acute etoposide stress-induced cell death

Because cancer cells are more tolerant to genotoxic stress than normal cells, and because we have shown that in the absence of

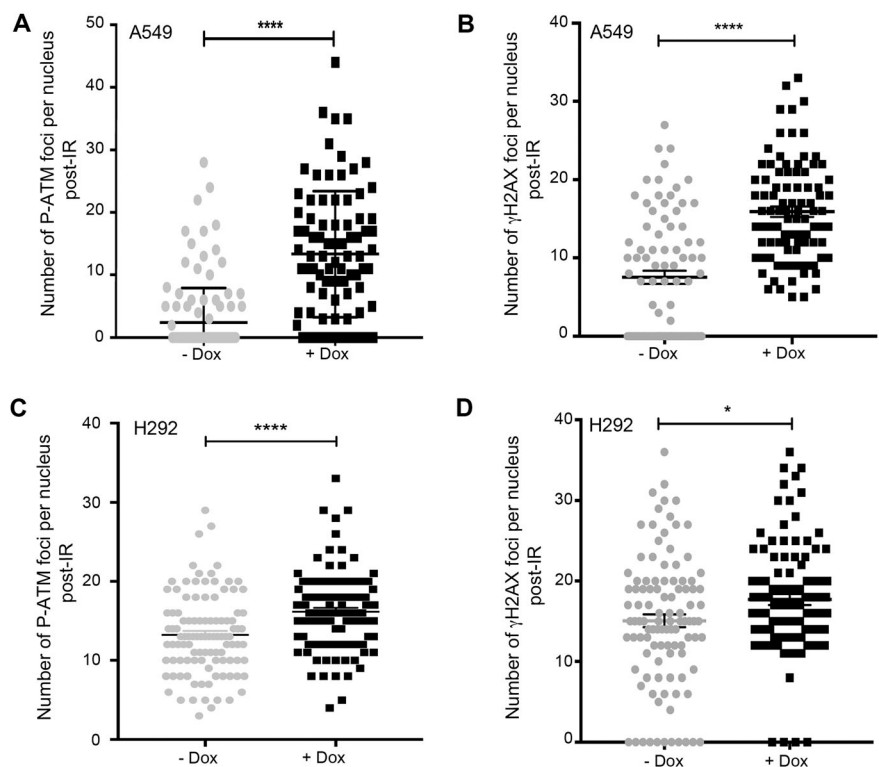

**Figure 3. Expression of NLRP3 in NSCLC cell lines results in increased ATM activation after the induction of DNA DSBs.**
**(A, B, C, D)** A549 (A, B) or H292 (C, D) cells stably expressing a doxycycline-inducible NLRP3 lentiviral vector (pSLIK-NLRP3) induced (+Dox) or not (−Dox) with 0.5 µg/ml doxycycline for 24 h were irradiated with 2 Gy, and P-ATM (A, C) and γH2AX (B, D) foci were assessed 1 h post-treatment (n ≥ 103 cells). ****P < 0.0001, *P < 0.05 (unpaired t test).

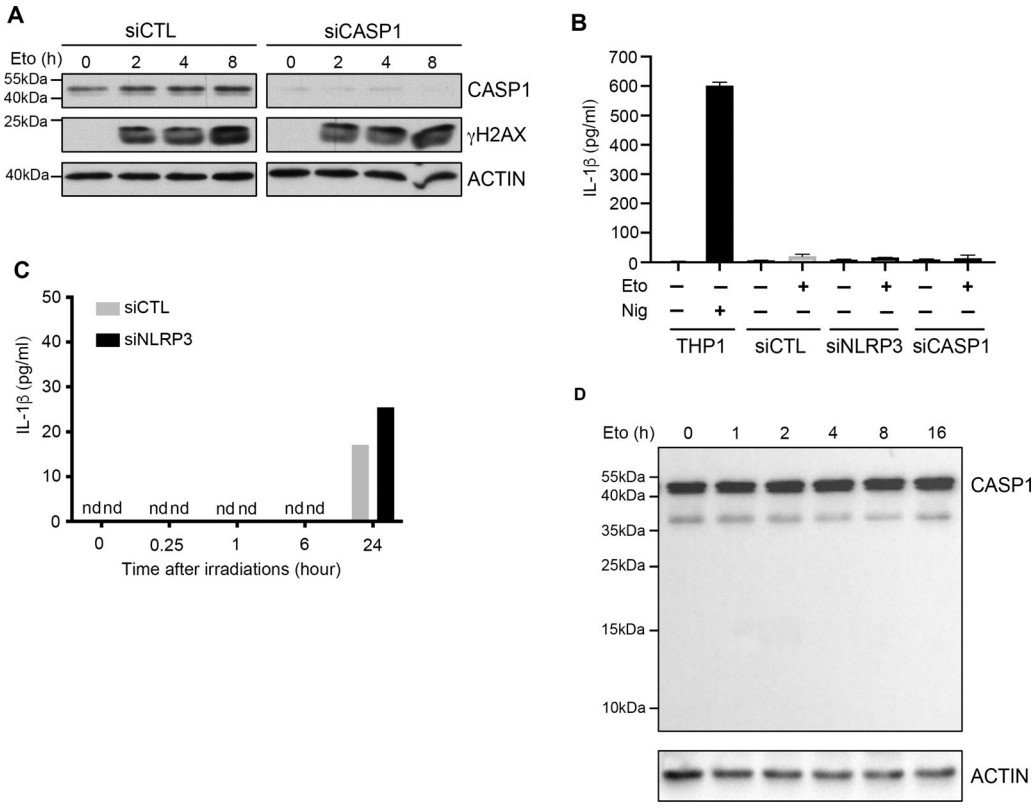

**Figure 4. NLRP3 controls the DDR in an inflammasome-independent manner.**
**(A)** HBEC3-KT control (siCTL) or caspase-1 siRNA (siCASP1) were treated with 100 $\mu$M of etoposide (Eto), and $\gamma$H2AX phosphorylation was monitored by immunoblotting. Actin served as a loading control. **(B)** HBEC3-KT transfected with either control, NLRP3, or caspase-1 (CASP1) siRNA were treated with Eto (100 $\mu$M) 8 h, and IL-1$\beta$ was quantified in cell supernatants by ELISA. Differentiated THP1 treated with nigericin for 3 h were used as a positive control for IL-1$\beta$ secretion. **(C)** IL-1$\beta$ secretion measurement at different time points in irradiated (2 Gy) HBEC3-KT transfected with control or NLRP3 siRNA using the Luminex assay. **(D)** HBEC3-KT cells were treated with 100 $\mu$M Eto over time, and caspase-1 cleavage was analyzed by immunoblotting. Actin served as a loading control. These data are one representative experiment out of two independent experiments. n.d, not detected.

NLRP3, ATM activation attenuation resulted in decreased P-p53 (Ser15), we next addressed whether NLRP3 loss would modulate resistance to Eto-induced apoptosis. To test this hypothesis, HBEC3-KT cells were transfected with siCTL or siNLRP3 and exposed to a lethal dose of Eto. In Eto-treated NLRP3-depleted HBEC3-KT cells, less caspase-3/7 activity was detected compared with siCTL, and an increase in the number of viable cells was observed (Fig 5A and B). As a control, ATM down-regulation by siRNA or ATM inactivation using the ATM inhibitor (ATMi) KU55933 conferred similar resistance to Eto (Fig S5A and B). The combination of siNLRP3 + ATMi was not more efficient in preventing apoptosis than siCTL+ATMi, demonstrating that ATM and NLRP3 are epistatic. These results suggest that decreased NLRP3 expression protects cells from Eto-induced apoptosis after DNA damage due to ATM pathway attenuation. Supporting this notion, the induction of *PUMA* and *NOXA/PMAIP1*, two p53 apoptosis effector genes, was significantly reduced in the absence of NLRP3 compared with control cells (Fig 5C and D) (Nakano & Vousden, 2001; Shibue et al, 2003). This response was specific to genotoxic stress as the induction of apoptosis via death receptor activation using a combination of TRAIL and MG132 did not result in impaired apoptosis in cells depleted for NLRP3 (Fig S5C). These data support a novel function for NLRP3 in the detection and subsequent response to DNA damage–induced genotoxic stress.

### NLRP3 forms a complex with ATM

Though NLRP3 is known as a cytoplasmic receptor, there have been reports that in Th2 lymphocytes NLPR3 was detected in the nucleus (Bruchard et al, 2015). Because ATM is activated on the formation of DSBs, we investigated whether in epithelial cells NLRP3 could also be localized in the nucleus. Cell fractionation experiments in HBEC3-KT revealed that endogenous NLRP3 was present in both the cytosolic and nuclear fractions under untreated or IR-treated conditions (Fig 6A), and live imaging in H292 also highlighted some mCherry-NLRP3 localized in the nucleus (Fig 6B), supporting earlier findings (Bruchard et al, 2015). Consistent with these observations, we found that NLRP3 interacted with IPO5 and XPO2, two proteins involved in protein shuttling between the nuclear and the cytosol compartments, which we had previously identified by mass spectrometry as potential NLRP3 interactors (Fig 6C and data not shown). To determine how NLRP3 regulates ATM activity, we tested the potential interaction between these two proteins. In HeLa cells, which do not express endogenous NLRP3, co-immunoprecipitation

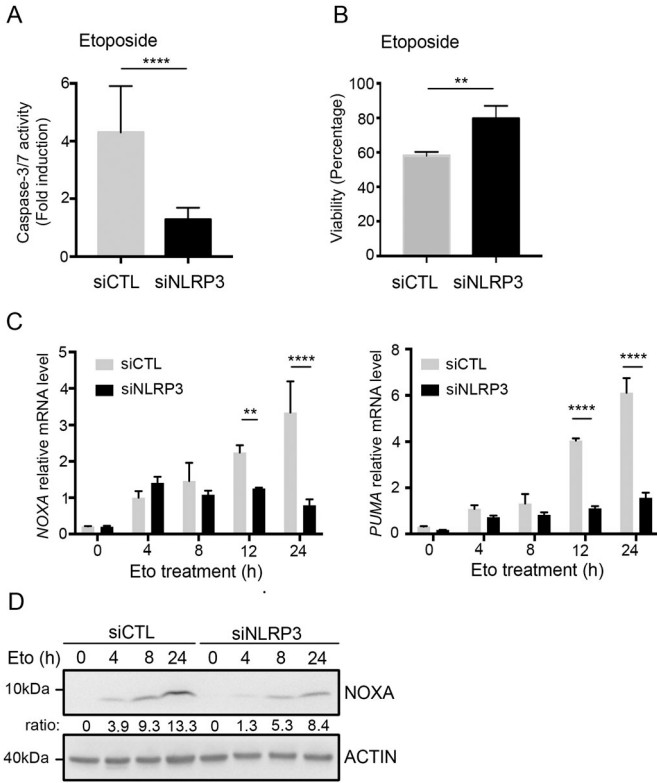

**Figure 5. The absence of NLRP3 confers resistance to acute genotoxic stress.**
**(A, B)** HBEC3-KT transfected with the indicated siRNA were treated with 50 µM etoposide (Eto) and (A) caspase-3/7 activity was measured by luminometry and (B) cell survival using the crystal violet cytotoxicity test. ****$P$ < 0.0001, **$P$ < 0.01 (unpaired $t$ test). Results are representative of three independent experiments. **(C)** *NOXA* and *PUMA* expressions were assessed in HBEC3-KT treated with 50 µM Eto at the indicated time points by Q-RT-PCR relative to *HPRT1* expression. **(D)** ****$P$ < 0.0001, **$P$ < 0.01 (multiple unpaired $t$ test) (D) NOXA expression was assessed by immunoblot. Actin was used as a loading control. The ratio between NOXA and actin is shown. Representative of two independent experiments.

(co-IP) of FLAG-ATM pulled down mCherry-NLRP3 (Fig 6D). We then determined the protein domains involved in the interaction by co-IP. We found that the HA-ATM kinase domain (aa 2566–3057) pulled down FLAG-NLRP3 (Fig 6E) and that FLAG-NLRP3-NACHT (domain present in neuronal apoptosis inhibitor protein, the major histocompatibility complex class II transactivator, HET-E, and TPI) and FLAG-NLRP3-LRR (leucin-rich repeats) NLRP3 domains bound to endogenous ATM but not the FLAG-NLRP3-PYD (pyrin) domain (Fig 6F) (Schroder & Tschopp, 2010; Shi et al, 2015a). Endogenous NEK7 was used as a positive interaction control, as it is a known partner of NLRP3.

We then tested whether the ATM-NLRP3 interaction was modulated by the formation of DSBs by treating HeLa cells with Eto and observed that the interaction was dissociated by DSB formation (Fig 6G). Using proximity ligation assay (PLA), we first validated in HeLa cells that FLAG-NLRP3 and endogenous ATM formed a complex (Fig S6A). Importantly, in HBEC3-KT and in MDA-MB-231 cells, we demonstrated that endogenous ATM and NLRP3 also interacted, and that the interaction decreased upon Eto treatment (Figs 6H and S6B). Collectively, these results establish that NLRP3

forms a complex with ATM, which is modulated by the formation of DSBs.

## Discussion

Functional links between innate immunity and DNA damage sensing pathways have been described. For instance, ATM was shown to be required in macrophages for optimal NLRP3 and NLRC4 inflammasome functions because its inactivation altered the ROS balance and therefore impaired inflammasome assembly (Erttmann et al, 2016). It was also suggested that DDR proteins such as RAD50 or BRCA1 are involved in the sensing of nucleic acids from viral pathogens in the cytosol (Roth et al, 2014; Dutta et al, 2015). Recently, cGAS was proposed to inhibit DNA repair, promoting genome instability and tumorigenesis (Liu et al, 2018; Jiang et al, 2019). However, little is known about the contribution of pattern recognition receptors to the sensing of stress like DNA damage. Here, our results highlight that NLRP3 is not only a major player in innate immunity but is also a key factor involved in the preservation of genome integrity through the modulation of the ATM signaling pathway in response to DSBs. We have demonstrated that NLRP3 is crucial to reach optimal ATM activation in response to DSB formation, and this function is inflammasome-independent. This previously unappreciated role for NLRP3 in the ATM pathway may be due to the fact that many common cellular models used in laboratories (such as MEF, HeLa, HEK293T, and A549) do not express NLRP3.

As previously reported, we found that NLRP3 is present in both cytosolic and nuclear compartments (Bruchard et al, 2015), consistent with its binding to XPO2 and IPO5. Using different approaches, we tried to force NLRP3 expression in the nucleus but were unsuccessful because of the toxicity of its overexpression (not shown). Thus, we speculated that NLRP3 translocation to the nucleus is tightly controlled to regulate ATM activation. DNA DSBs induce ATM auto-phosphorylation, resulting in its monomerization and activation (Bakkenist & Kastan, 2003). Mechanistically, we uncovered that NLRP3 forms a complex with ATM that dissociated after DSB induction, and NLRP3 KD resulted in decreased levels of ATM auto-phosphorylation, a marker of ATM activation, and the subsequent activation of downstream effectors. NLRP3 is an AAA+ ATPase and proteins of this family are involved in DNA repair and protein stability (Snider et al, 2008). Therefore, we propose that NLRP3 may act as a chaperone for ATM.

As a consequence of ATM activity attenuation in NLRP3-depleted cells, less Ser15 p53 phosphorylation was detected, which is important for p53 transcriptional activity in response to DNA damage. Therefore, NLRP3 KD conferred cell resistance to acute Eto-induced apoptosis due to impaired *NOXA* and *PUMA* gene expressions (Shibue et al, 2006; Loughery et al, 2014). R. Flavell's group demonstrated in mouse models that the severe damage caused to the gastrointestinal tract and the hematopoietic system in response to whole body γ-irradiation is due to the activation of the AIM2 inflammasome by DSBs, which causes massive cell death by caspase-1–dependent pyroptosis in these fast-renewing tissues (Hu et al, 2016). Our results contrast with these findings as we neither observe caspase-1 activation nor IL-1β production upon DNA DSB induction in our models.

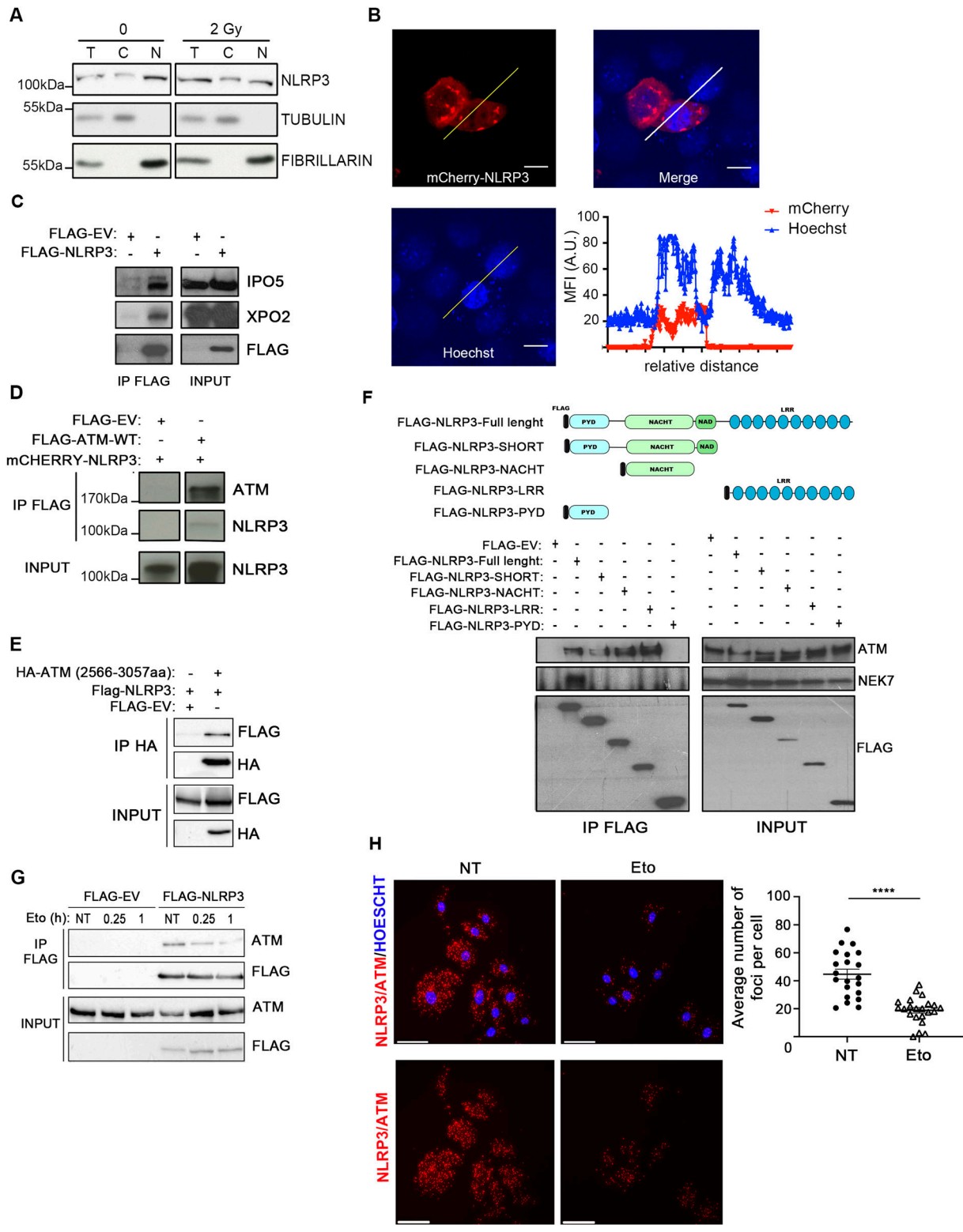

**Figure 6. NLRP3 forms a complex with ATM.**
**(A, C)** HBEC3-KT untreated (0) or irradiated (2 Gy) were fractionated, and NLRP3 expression was analyzed by immunoblot from the cytosolic (C) and nuclear (N) fractions. Tubulin was used as a marker of the cytosolic fraction and fibrillarin of the nuclear fraction. T is total lysate. Representative of three independent experiments. **(B)** NLRP3 was detected in the cytosolic and nuclear compartments in live confocal imaging of mCherry-NLRP3 transfected H292 cells. The graph displays mean fluorescence intensity (arbitrary unit) according to the relative distance. Vital Hoechst was used to stain nuclei. Scale bar 10 μm (×100). **(C, D)** Immunoblot of HeLa cells transfected with FLAGempty vector (EV) or FLAG-NLRP3 (INPUT) after FLAG immunoprecipitation (IP FLAG) (D) immunoblot of HeLa cells transfected with indicated vectors after IP

These observations would suggest that tissue and species-specific differences may exist that warrant further investigations.

The DDR is known to be a barrier to cancer in the early phases of tumorigenesis (Bartkova et al, 2005; Lantuejoul et al, 2010). *TP53* is frequently mutated in cancer, and the ATM pathway is down-regulated in many solid tumors: 11% of NSCLC carry somatic mutations in *ATM* and 41% of LUAD and 75% of BC have reduced ATM protein expression (Angèle et al, 2003; Ding et al, 2008; Cancer Genome Atlas Research Network, 2012; Imielinski et al, 2012; Rondeau et al, 2015; Villaruz et al, 2016). Our data unraveled that in NSCLC and BC, NLRP3 expression was significantly lower than normal tissues, and its expression was positively correlated with genome stability and patient outcome. Interestingly, a recent bioinformatic analysis also reported decreased NLRP3 expression in a pan-cancer analysis including NSCLC (Ju et al, 2021). Thus, we propose that the down-regulation of *NLRP3* expression during malignant transformation may represent an additional mechanism to attenuate ATM and p53 signaling pathways, allowing cells to survive genotoxic stress despite the presence of genome alterations. Hence, loss of NLRP3 and the subsequent impairment of the ATM pathway could be an event allowing cells to progress toward malignancy. In the future, it will be interesting to define at the molecular level the link between NLRP3 and genome stability.

In summary, we reveal an unexpected role for NLRP3 in DNA damage sensing by demonstrating that loss of NLRP3 impairs the ATM pathway. Our study opens new perspectives for innate immune receptors in preventing progression toward malignancy.

# Materials and Methods

### Cell culture and treatments

Human broncho-epithelial cells HBEC3-KT (immortalized with hTERT and CDK4), HBEC3-KTRL53 (HBEC3-KT with stable expression of *TP53* shRNA), HBEC3-ET (immortalized with E6/E7), HCC15*, HCC366*, HCC4017*, and HCC461 were obtained from J. Minna. NSCLC cells: NCI-H1703 (H1703), NCI-H292 (H292), NCI-H520 (H520), NCI-H661* (H661), NCI-H358* (H358), NCI-H1792 (H1792), NCI-H441 (H441), SK-MES-1, HEK293T, MDA-MB231, and HeLa cells were supplied by ATCC, and the NSCLC cell line A549 was from the PHE culture collection. * cell lines carrying NLRP3 somatic mutation. HMEC-hTERT, HMLE, and HMLER were obtained from AP Morel (Morel et al, 2008), and HMEC-hTERT was cultured in 1:1 DMEM/Ham F12 medium (Gibco-Thermo Fisher Scientific), supplemented with 1% GlutaMAX (Gibco-Thermo Fisher Scientific), 10 ng/ml human EGF (PromoCell), 0.5 mg/ml hydrocortisone (Sigma-Aldrich), and 10 mg/ml insulin (Actrapid). HBEC3-KT cells were cultured in keratinocyte-serum free medium (Invitrogen), supplemented with 12.5 mg of bovine pituitary extract (Invitrogen) and 100 ng of epidermal growth factor (Invitrogen). H1792, A549, HCC15, HCC366, HCC4017, HCC461, and H441 were cultured in RPMI medium, supplemented with 10% FBS and 1% penicillin/streptomycin, H1703, H292, H520, H358, and H661 in RPMI, 10% FBS, 1 mM sodium pyruvate, 1 mM HEPES, and 1% penicillin/streptomycin and SK-MES-1 in RPMI, 10% FBS, 1 mM sodium pyruvate, 1 mM non-essential amino acids, and 1% penicillin/streptomycin (Invitrogen). HeLa cells were cultured in DMEM 4.5 g/l of glucose, 10% FCS, and 1% penicillin/streptomycin. Treated cells received Eto (TEVA santé 100 mg/5 ml or Selleckchem) or γ-ray treatment 36–48 h post siRNA transfection. For inflammasome activation, THP1 cells were cultured and differentiated, as described in Pétrilli et al (2007b) and treated with nigericin (10 µM, 3 h) (Sigma-Aldrich). Z-VAD-fmk and z-YVAD-fmk were from Enzo Life Science.

### Cell transfection

HBEC3-KT were seeded at $1.5 \times 10^5$ cells per well in six-well plates and transfected with non-targeting, NLRP3, CASP1, or ATM siRNA (ON-TARGETplus SMARTpool; Dharmacon) using HiPerFect Transfection Reagent (QIAGEN) or INTERFERin (Polyplus transfection) following the manufacturer's instructions. MDA-MB-231 and HMEC-hTERT were transfected with INTERFERin (Polyplus transfection). HeLa cells were transfected with Lipofectamine 2000 (Invitrogen). H292 or A549 cells were transfected using polyethylenimine (PEI; Polyscience Inc.) according to the manufacturer's instructions. pCR3-FLAG-NLRP3–based vectors (FL, SHORT, PYD, NACHT, and LRR) were a gift from the Tschopp laboratory. pcDNA3.1-FLAG-His-ATM was a gift from Michael Kastan (31985; Addgene) (Canman et al, 1998), mCherry-NLRP3 was cloned in pcDNA3.1. DNA constructs for pcDNA3-ATM kinase domain (7,699–9,171 bp) was generated by in vitro DNA synthesis by GeneCust (Dudelang). shRNAs with hygromycin selection were from GeneCopoeia, *NLRP3*: forward: 5′-TAATACGACTCACTATAGGG-3′ reverse: 5′-CTGGAATAGCTCAGAGGC-3′; control forward: 5′-TAATACGACTCAC TATAGGG-3′ reverse: 5′-CTGGAATAGCTCAGAGGC-3′.

### Irradiation

Cells were irradiated using a 6-MeV γ-ray clinical irradiator (SL 15 Phillips) at the Centre Léon Bérard, with a dose rate of 6 Gy.min⁻¹ to obtain the required dose.

### ROS measurement

For intracellular ROS staining, HBEC3-KT cells were incubated with 1 µM of 2′–7′-dichlorofluorescin diacetate (CM-H2DCFDA; Invitrogen) for 30 min at 37°C. For a positive control, cells were

---

FLAG. Representative of three independent experiments. **(E)** Immunoblot of HeLa cells transfected with a Flag-EV, FLAG-NLRP3, and HA-ATM kinase domains (2,566–3,057) after IP HA. Representative of two independent experiments. **(F)** Different FLAG-tagged NLRP3 domain constructs were transfected into HeLa cells, and FLAG-proteins were immunoprecipitated. Pulled-down proteins were analyzed by immunoblot. Representative of two independent experiments. **(G)**. Immunoblots of HeLa cells transfected with Flag-EV or FLAG-NLRP3 after FLAG IP in the presence or absence of etoposide (Eto) (50 µM) for the indicated time points. The presence of endogenous ATM was analyzed in the INPUT panel and in the IP FLAG. **(H)** Representative of three independent experiments (H) proximity ligation assay was performed in HBEC3-KT cells treated or not with Eto 50 µM for 2 h (number of nuclei analyzed 286≤ n ≥302) using anti-ATM and anti-NLRP3 (×40). Representative of two independent experiments. DAPI (blue) was used to stain nuclei. Scale bars 50 µm. Signal quantification is shown on the graph on the right panel. NT, not treated. ****$P < 0.0001$ (unpaired *t* test).

pretreated with 5 µM of ATMi (KU55933; Selleckchem) for 5 h before staining. Stained cells were collected and analyzed on a BD FACSCalibur, and data were analyzed using FlowJo software.

## Mathematical modeling

ATM dynamics was modeled using the following ordinary differential equation:

$$\frac{dA}{dt} = k_{IR}1_{t \in [IRexposure]} - k_{inact}A,$$

where A is the concentration of activated ATM molecules (expressed in number of foci/cell), $k_{IR}$ is the activation rate (in number of foci/cell. $h^{-1}$), only present during irradiation, and $k_{inact}$ is the inactivation rate (in $h^{-1}$). A was set at zero at t0. This model assumed that ATM molecules were in excess compared with the activated proportion.

The model is fitted to experimental data by estimating the optimum values for $k_{IR}$ and $k_{inact}$ for each condition, siCTL or siNLRP3, using a weighted least square approach (Hill et al, 2020). For hypothesis A, that is, NLRP3 enhances ATM activation, $k_{IR}$ is assumed to be different for both conditions whereas $k_{inact}$ is taken as identical. For hypothesis B, that is, NLRP3 inhibits ATM deactivation, $k_{inact}$ is assumed to vary between siCTL and siNLRP3 conditions, and $k_{IR}$ is assumed to remain identical. All computations were done in MATLAB (Mathworks).

## Generation of cells stably expressing NLRP3

The human NLRP3 cDNA was inserted into the lentiviral vector pSLIKneo (Addgene) containing a TET-ON promoter using the Gateway recombination system (Invitrogen). Sequences of the Gateway shuttle vectors are available upon request. Empty pSLIK vector (without the ccdB containing Gateway recombination cassette) was produced by partial digestion with Xba1 and Xho1 after religation. Viral particles were produced by transfecting HEK293T cells with the lentiviral vectors and a second-generation packaging system (psPAX and pMD2.G plasmids). Lentiviral supernatant was collected 48 h after transfection, filtered through a 0.45-µM filter, supplemented with 6 µg/ml polybrene (Sigma-Aldrich), and added to cell lines. NCI-H292 (H292) and A549 cells were either transduced with the empty pSLIK control vector or the NLRP3-containing vector. NLRP3 expression was induced by adding 0.5 µg/ml doxycycline (Takara Bio) to the cell culture medium.

## Western blotting

Cell pellets were lysed in Laemmli buffer ×2 (Tris–HCl 0.5 M, pH 6.8; 0.5 M DTE; 0.5% SDS) and protein concentrations determined using the Bradford reagent (Bio-Rad). Protein extracts were separated on SDS–PAGE (8% or 15% or 4–15% gradient [vol/vol]) gels. Gels were transferred onto nitrocellulose membranes (GE HealthCare and Bio-Rad) for immunoblotting with the following antibodies: anti-NLRP3 (Cryo-2, 1:1,000) and anti-caspase-1 (Bally-1, 1:1,000) from AdipoGen, anti-ASC (1:2,000) from Enzo Life Science, and anti-γH2AX (JBW301, 1:1,000), anti-P-Ser15-p53 (1:1,000), and anti-ATM Ser1981 (10H11.E12, 1:1,000) from Millipore. Anti-P-KAP1 Ser824 (1:1,000), anti-

KAP1 (1:1,000), and anti-NEK7 (A302-684A, 1:1,000) from Bethyl Laboratories, anti-p53 (clone DO7 1:2,000) and anti-NOXA (114C307, 1:1,000) from Santa Cruz; anti-ATM (#ab32420, 1:1,000), anti-fibrillarin (ab4566, 1:1,000), and anti-GAPDH (ab9484, 1:1,000) from Abcam; anti-FLAG (F7425 1:5,000) and anti-α-tubulin (clone B-5-1-2 1:1,000) from Sigma-Aldrich; anti-XPO2 (GTX102005 1:1,000) and anti-IPO5 (GTX114515 1:1,000) from Genetex, and anti-actin (C4, 1:100,000) from MP Biomedical. The Fiji and Image Laboratory (Bio-Rad) programs were used for densitometric analysis of immunoblots, and the quantified results were normalized as indicated in the figure legends.

## Cell fractionation

HBEC3-KT were fractionated by adapting the method described by Hacot et al (2010). The $MgCl_2$ concentration used for the hypotonic buffer was 0.75 mM. Equal amounts of proteins were run by SDS–PAGE.

## Immunofluorescence

Cells were plated onto sterile glass coverslips and fixed with PBS-PFA 4% (wt/vol) for 15 min at RT and washed twice in PBS. Cells were permeabilized with lysis buffer (sucrose 300 mM, $MgCl_2$ 3 mM, Tris, pH 7.0, 20 mM, NaCl 50 mM, and Triton X-100 0.5%) for 3–5 min at RT under slow agitation. The following antibodies were diluted in PBS-BSA 4% and applied to the coverslips for 40 min at 37°C: anti-γH2AX (JBW301, 1:800) and P-ATM Ser1981 (10H11.E12, 1:200) from Millipore. Cells were then incubated with Alexa-Fluor 488-conjugated anti-mouse or Alexa-Fluor 555-conjugated anti-rabbit (1:800; Life Technologies) for 20 min at 37°C and then in Hoechst 33342 (500 ng/ml in PBS) for 10 min at RT. Fluorescence microscopy pictures were taken using a Nikon Eclipse Ni-E microscope, or a confocal Zeiss LSM 780. The Fiji program was used to analyze fluorescence intensity and number of foci per nuclei.

## Live imaging

mCherry-NLRP3-transfected H292 was imaged using a confocal spinning disk inverted microscope (Leica, Yokogawa). Vital Hoechst 33342 was used at 500 ng/ml. The Fiji program was used to analyze fluorescence intensity.

## Co-IP

HeLa cells were lysed in the following buffer: Tris–HCl 100 mM, pH 8.0, 10 mM $MgCl_2$, 90 mM NaCl, 0.1% Triton X-100, and Complete tablet (Roche) (Qu et al, 2012). Immunoprecipitation was performed using M2-agarose beads (A2220; Sigma-Aldrich) or anti-HA-agarose beads (A2095; Sigma-Aldrich) overnight at 4°C.

## Proximity ligation assay

HBEC3-KT, MDA-MB231, and HeLa cells were seeded onto glass coverslips, transfected as previously described and processed as described by the manufacturer's protocol (Duolink PLA Technology, Sigma-Aldrich) using NLRP3 2 µg/ml (ABF23; Millipore) and ATM 2 µg/ml (2C1; Abcam) antibodies. DAPI was used to stain nuclei.

Quantification was carried out using the macro published by Poulard et al (2019).

### IL-1β luminex assay and ELISA

IL-1β levels in cell supernatants were analyzed using either R&D systems DuoSet ELISA or the Magnetic Luminex Screening Assay according to the manufacturer's protocol (R&D Systems) and analyzed using the Bio-Plex-200 from Bio-Rad.

### Quantitative reverse transcription PCR

RNA was extracted using the NucleoSpin RNA kit (Macherey Nagel). 500 ng to 1 μg of RNA were reverse transcribed using SuperScript II reverse transcriptase and oligo(dT) primers (Life technologies) in the presence of RNAs inhibitor (Promega). cDNAs were quantified by real-time PCR using a SYBR Green PCR Master Mix (Applied Biosystems or Bio-Rad) on an ABI Prism 7000 (Applied Biosystems) or CFX Connect Real-Time system (Bio-Rad). Sequences of the primers NLRP3 forward 5′-GAAGAAAGATTACCGTAAGAAGTACA-GAAA; reverse 5′-CGTTTGTTGAGGCTCACACTCT; ESD 5′-TTAGATGGA-CAGTTAC TCCCTGATAA; reverse 5′-GGTTGCAATGAAGTAGTAGCTATGAT; HPRT1 forward 5′-CATTATGCTGAGGATTTGGAAAGG; reverse 5′-TGTAGCCCTCTGTGTGCTCAAG; CBP forward 5′-CGGCTGTTTAACTTCGCTTC; reverse 5′-CACACGCCAAGAAACAGTGA. NOXA forward 5′-GGA-GATGCCTGGGAAGAAG; reverse 5′-CCTGAGTTGAGTAGCACACTCG; PUMA forward 5′-CCTGGAGGGTCCTGTACAATCTCAT; reverse 5′-GTATGCTA-CATGGTGCAGAGAAAG; and ACTIN forward 5′-AGCACTGTGTTGGCGTA-CAG; reverse 5′-TCCCTGGAGAAGAGCTACGA.

### Caspase-3/7 assay

Cells were cultured in 96-well plates and treated with 50 μM of Eto for 12 h or with 200 ng/ml TRAIL (Peprotech) and 1 mM MG132 (Sigma-Aldrich). Caspase 3/7 activity was assessed using the Caspase-Glo 3/7 assay reagent (Promega) following the manufacturer's instructions. KU55933 was added 5 h before treatment at 5 μM, DMSO was added to all other wells. The luminescence was measured using a TECAN Infinite M200PRO luminometer microplate reader. To normalize the results, a second plate was stained with crystal violet and analyzed as described in the Crystal violet cytotoxicity assay section below.

### Crystal violet cytotoxicity assay

Cells were stained with 0.5% crystal violet (Sigma-Aldrich Corp.) in 30% methanol for 20 min at RT. Cells were lysed in a 1% SDS (Sigma-Aldrich Corp.) solution. The absorbance of the solution was measured using a TECAN Infinite M200PRO microplate reader at a wavelength of 595 nm.

### Soft agar assay

The test was performed as previously described by Morel et al (2008). Pictures were taken using binocular loop or under a bright field microscope.

### Tissues from NSCLC patients

Frozen lung tumor tissues from non-treated patients were obtained from the Biological Resource Centre in Grenoble (Centre de Ressources Biologiques du CHU de Grenoble, CRB CHU GA, ISO9001:2015, and NFS 96900) n°BB-0033-00069. All tissue collections are registered to the French ministry of higher education and research according to the public health code law L. 1,243-4 (AC-2007-23; AC-2014-2094; AC 2019–3627) and collected with patient approval. The CRB RNA was extracted from regions containing mainly malignant cells using the AllPrep RNA/DNA kit from QIAGEN.

### TCGA data analysis

RNAseqV2 data of LUAD or BC (PAM50 subtypes were taken from Knijnenburg et al [2018]) and corresponding clinical data were downloaded from The Cancer Genome Atlas data portal (https://gdc.cancer.gov). The Cox regression model was used to estimate hazard ratio (HR) and 95% confidence intervals (CIs) for OS and PFI. NLRP3 expression was dichotomized using the methodology of Lausen et al via the maxstat library or each variable, the first group containing at least 30% of the population was used (Lausen & Schumacher, 1992). Survival probabilities were estimated using the Kaplan–Meier method, and survival curves were compared using the log-rank test. For the comparison with NLRP3 expression in pan-cancer or breast, colon, pancreas, and prostate tissues, differences between the median values of two groups were compared using the Welch's t test on the UCSC Zena browser (Goldman et al, 2020).

For TCGA breast and LUAD cancer cohorts, clinical, expression, and the fraction of genome altered (FGA%) according to NLRP3 expression data were obtained from TCGA database (TCGAPanCancer Atlas) using cBioportal (http://www.cbioportal.org/). High, intermediate, and low samples were defined for gene expression analyses.

### Statistical analysis

Statistical analysis of the experimental data was performed using GraphPad. Unpaired group comparisons were done using two-tailed t test, or Mann–Whitney test. Concerning TCGA data analysis for patients OS and PFI, statistical analyses were performed using R software (http://www.R-project.org/) (version 4.0.2, accessed on 22 June 2020), and graphs were drawn using GraphPad Prism version 7.03.

## Supplementary Information

## Acknowledgments

We thank John Minna for sharing the HBEC3-KT and NSCLC cells, Dr Puisieux and Dr Morel for the HMEC model, Dr Foray's team and Marine Malfroy for technical help with irradiation protocols, and Pascale Bertrand for helpful discussions. We thank Christophe Vanbelle (CRCL Imaging platform) and

Christophe Chamot of LyMIC-PLATIM from the SFR biosciences (UAR3444/CNRS, US8/Inserm, ENS de Lyon, UCBL) for excellent technical help, and Brigitte Manship for manuscript editing. M Bodnar-Wachtel was supported by the ANRT, V Petrilli by the plan Cancer, Ligue Contre le Cancer Comité de l'Ain, the ARC foundation, and the Fondation pour la Recherche Médicale DEQ20170336744, A-L Huber by the ARC foundation, and by the European Union's Horizon 2020 research and innovation program under the Marie Skłodowska-Curie grant agreement No 751216, N Goutagny was supported by the CLARA, and B Guey by the ARC foundation. BF Py was supported by the ERC-2013-CoG_616986. D Burlet was supported by the Institut Convergence Institut Convergence PLAsCAN, ANR-17-CONV-0002. Y Couté acknowledges the support of the Agence Nationale de la Recherche under projects ProFI (Proteomics French Infrastructure, ANR-10-INBS-08) and GRAL, a program from the Chemistry Biology Health (CBH), Graduate School of University Grenoble Alpes (ANR-17-EURE-0003). Graphical abstract was created with Biorender.com.

## Author Contributions

M Bodnar-Wachtel: conceptualization, formal analysis, investigation, methodology, and writing—original draft.
A-L Huber: conceptualization, formal analysis, investigation, methodology, and writing—original draft, review, and editing.
J Gorry: conceptualization, investigation, and writing—original draft.
S Hacot: formal analysis, investigation, methodology, and writing—original draft.
D Burlet: investigation and methodology.
L Gérossier: investigation and methodology.
B Guey: investigation and methodology.
N Goutagny: methodology.
B Bartosch: methodology.
E Ballot: investigation and methodology.
J Lecuelle: investigation and methodology.
C Truntzer: investigation and methodology.
F Ghiringhelli: methodology.
BF Py: methodology.
Y Couté: investigation and methodology.
A Ballesta: resources, investigation, and methodology.
S Lantuejoul: resources and methodology.
J Hall: conceptualization, investigation, methodology, and writing—original draft, review, and editing.
A Tissier: investigation, methodology, and writing—review and editing.
V Petrilli: conceptualization, formal analysis, supervision, funding acquisition, investigation, methodology, project administration, and writing—original draft, review, and editing.

## Conflict of Interest Statement

The authors declare that they have no conflict of interest.

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
