## [Reviewer comments · Life Science Alliance]

Life Science Alliance

Inflammasome-independent NLRP3 function enforces ATM activity in response to genotoxic stress.

Virginie Petrilli, Mélanie Bodnar-Wachtel, Anne-Laure Huber, Julie Gorry, Sabine Hacot, Delphine Burlet, Laetitia Gérossier, Baptiste Guey, Nadège Goutagny, Birke Bartosch, Elise Ballot, Julie Lecuelle, Caroline Truntzer, François Ghiringhelli, Bénédicte Py, Yohann Couté, Annabelle Ballesta, Sylvie Lantuejoul, Janet Hall, and Agnès Tissier

DOI: <https://doi.org/10.26508/lsa.202201494>

Corresponding author(s): Virginie Petrilli, Cancer Center of Lyon

Review Timeline:

Submission Date:	2022-04-21
Editorial Decision:	2022-06-07
Revision Received:	2022-12-15
Editorial Decision:	2023-01-09
Revision Received:	2023-01-20
Accepted:	2023-01-20

Scientific Editor: Novella Guidi

Transaction Report:

June 7, 2022

Re: Life Science Alliance manuscript #LSA-2022-01494-T

Virginie Petrilli
Centre de Recherche en Cancérologie de Lyon
FRANCE

Dear Dr. Petrilli,

Thank you for submitting your manuscript entitled "Inflammasome-independent NLRP3 function enforces ATM activity in response to genotoxic stress." to Life Science Alliance. The manuscript was assessed by expert reviewers, whose comments are appended to this letter. We invite you to submit a revised manuscript addressing the Reviewer comments.

Thank you for this interesting contribution to Life Science Alliance. We are looking forward to receiving your revised manuscript.

Sincerely,

B. MANUSCRIPT ORGANIZATION AND FORMATTING:

Reviewer #1 (Comments to the Authors (Required)):

"Inflammasome-independent NLRP3 function enforces ATM activity in response to genotoxic stress" By Bodnar-Wachtel et; al.

In this manuscript, the authors reveal a novel actor of the DNA damage signalling at the ATM level (early step of the DNA damage response): NLRP3, which is known to acts in the inflammasome assembly. This corresponds to a novel function of NLRP3 because the activation of ATM by NLRP3 appeared to be independent of the inflammasome functions. The authors show the physical interactions between ATM and NLRP3. Then they studied the consequences on signalling and on cell viability. They also show the downregulation of NLRP3 (and the bad prognostic of such misregulation) in cancer. The data are new, unexpected and interesting; they reveal a novel actor of the DNA damage response. However, the authors should consider the following comments prior to publication.

- The molecular mechanism allowing NLRP3 to optimise ATM signalling is poorly documented and not discussed. The authors show that NLRP3 and ATL interact, but that DSBs favour their dissociation, which is counter-intuitive. The authors should document the molecular mechanisms of how NLRP3 act, or at least discussed it or propose speculations in the discussion.

- Deficiency in ATM leads to severe radiation sensitivity. AT-syndrome is a paradigm of radiation sensitivity. Because suppression of NLRP3 leads to non-efficient ATM activation, this should lead to radiation sensitivity (unless there is another additional mysterious role). In contrast the author found resistance to etoposide. This should be discussed/comment this paradox.

- The authors start with the analysis of NLRP3 in cancer prior to studying the ATM connection, which correspond to the central topic of the manuscript. However, the link between these two parts is not really clearly explained. The authors might want to consider starting with the ATM connection analyses ,then to switch the studies in cancer at the end (they did this organization in the discussion).

There are also several points to consider: - what is the part of inflammasome defect and of DDR defect in the oncogenesis? For instance, what are the expressions of other inflammasome genes (ASC, caspase1) in the same cohorts? Are they co-deregulated with NLRP3, or in contrast, they are deregulated in different tumours? What is the consequence of the other inflammasome genes in cancer? This also should be discussed.

- lines 211-215: The fact that that inhibition of both NLRP3 and ATM is more sensitive than single inactivation suggest that they do not act in an epistatic way? This would be in contradiction with the rest of the manuscript.

- line280. (Hu et al, 2015); I think this is not the correct citation, or Flavall's citation is missing.

• - Figure 1C. There is a problem with the loading control; there is more actin signal that can account for the fact that there is also more NLRP3 in HMEC-tert cell than in the other cell lines.

Reviewer #2 (Comments to the Authors (Required)):

Understanding the ATM regulation is of crucial interest, with direct implication for cancer research/therapy. Presented manuscript aspires to bring another layer of complexity to ATM regulation: suggests functional link between native immunity component NLRP3 (constituent of inflammasome) and activation of ATM in response to DNA damage.

Using two cancer models, namely broncho-epithelial cells and breast cancer cells, authors showed decreased ATM activation in absence of NLRP3 after etoposide or ionizing radiation DNA damage-inducing treatment. Analysis of downstream effectors of ATM (including phosphorylated H2AX, Chk2, p53,) confirmed the NLRP3 requirement. Furthermore, in absence of NLRP3 significantly lower number of gH2AX and P-ATM foci was detected, in accordance with ATM activation impairment. Evidence provided here is convincing, further corroborated using regulatable expression of NLRP3 in non-small cell lung cancer and breast cancer cell lines.

Next, the authors addressed whether NLRP3-coupled inflammasome activity is involved in response to DNA damage/repair. Based on the data, it seems that NLRP3 involvement in ATM activation is independent of the functional inflammasome activity and that DNA damage induced does not activate inflammasome effectors. Conclusion points to the new, inflammasome-independent role of NLRP3.

Regarding the attenuated ATM activity resulting in decreased P-p53 in absence of NLRP3, the apoptosis inhibition was investigated, and confirmed. However, besides cell death, other downstream effects of ATM signalling impairment could have been addressed. Taking into account decrease in DNA damage signaling and lowered number of DNA repair foci resulting from NLRP3 absence, overall impairment of DNA repair kinetics (and effectivity) could be expected. Addressing the DNA repair characteristics of induced DNA damage in the absence of NLRP3 may provide direct evidence of functional consequences of NLRP3 downregulation. Long-term effects of DNA repair impairment should be reflected also on the genome in/stability of NLRP3-compromised cells.

An effort to provide mechanistic clue on NLRP3-ATM interaction is the subject of the final part of the experimental work. The pull-down data showing the ATM-NLRP3 interaction seem to be trustworthy. Less convincing I found the results of Proximity Ligation Assay, as majority of foci (see Fig.6H, not treated sample on the left) are located outside the nuclei. Conclusion based on these data Moreover, the role of cytoplasmic ATM is considered to be different from nuclear ATM function.

Altogether, manuscript brings significant claims on interesting topic and presents valuable contribution to the field. To improve the quality of the manuscript I suggest following points to be addressed:

1. To strengthen the conclusion of the paper, addressing the possible DNA repair impairment would be beneficial. One option is to bring direct evidence of changes in DNA damage removal/persistence in NLRP3-compromised cells (comet assay may be the tool), another way is to provide the data supporting the evidence of increased genomic instability as a consequence of long-term ATM activity impairment caused by NLRP3 downregulation.
2. Regarding the quantification of foci in Proximity Ligation Assay (Fig.6H), the question is whether the foci were counted only in DAPI-stained nuclear area, or in whole cell including cytoplasm. The immunofluorescence image displaying only the red channel for the same area should be included in the final Fig.6H to provide the evidence of nuclear foci, preferably in higher magnification. Currently presented merged images in Fig.6H do not allow for verification of nuclear foci presence.
3. The experimental part of the manuscript is focused on 2 types of cancer. To help appreciate the scale of NLRP3 dysregulation in cancer pathogenesis, the information on another types of cancer associated with NLRP3 could be included in the paper.

Reviewer #1 (Comments to the Authors (Required)):

"Inflammasome-independent NLRP3 function enforces ATM activity in response to genotoxic stress" By Bodnar-Wachtel et; al.

In this manuscript, the authors reveal a novel actor of the DNA damage signalling at the ATM level (early step of the DNA damage response): NLRP3, which is known to acts in the inflammasome assembly. This corresponds to a novel function of NLRP3 because the activation of ATM by NLRP3 appeared to be independent of the inflammasome functions. The authors show the physical interactions between ATM and NLRP3. Then they studied the consequences on signalling and on cell viability. They also show the downregulation of NLRP3 (and the bad prognostic of such misregulation) in cancer. The data are new, unexpected and interesting; they reveal a novel actor of the DNA damage response. However, the authors should consider the following comments prior to publication.

We thank the reviewer for the positive comments on the novelty of our findings. We have addressed below the comments that were raised.

- The molecular mechanism allowing NLRP3 to optimise ATM signalling is poorly documented and not discussed. The authors show that NLRP3 and ATL interact, but that DSBs favour their dissociation, which is counter-intuitive. The authors should document the molecular mechanisms of how NLRP3 act, or at least discussed it or propose speculations in the discussion.

Despite different approaches to elucidate how NLRP3 optimizes ATM activation, the exact mechanism has not yet been pinpointed. Our current hypothesis is that NLRP3 traffics to the nucleus with ATM under physiological conditions. We tried to target NLRP3 to the nucleus using tamoxifen inducible ER domain or triple Nuclear Localisation Signal, and both were highly toxic to the cells. Because we could not over-express these nuclear mutants, we are currently using PLA to detect ATM/NLRP3 in the nucleus. Our preliminary results indicate that this shuttling is following a rhythmic pattern. But these observations need to be strengthened and will be the subject of a future manuscript. Based on our present data, we believe that the shuttling of NLRP3 to the nucleus is tightly controlled and involved in the activation of ATM by DNA double strand breaks (DSBs). Since NLRP3 is an AAA+ ATPase it may act as a chaperone for ATM with the sensing of DSBs by ATM promoting the release of NLRP3 from the ATM/NLRP3 complex. We have added this information in the manuscript discussion.

- Deficiency in ATM leads to severe radiation sensitivity. AT-syndrome is a paradigm of radiation sensitivity. Because suppression of NLRP3 leads to non-efficient ATM activation, this should lead to radiation sensitivity (unless there is another additional

mysterious role). In contrast the author found resistance to etoposide. This should be discussed/comment this paradox.

We agree that an expected phenotype would be radio-sensitivity. Unfortunately, HBEC3-KT do not clone, thus we were not able to test this using classic sensitivity approaches. However, we assessed the survival of NLRP3 knock-down (KD) to low dose (1 μ M) and short exposure (3h) of etoposide. As shown in the figure below, NLRP3 KD cells displayed an increase in sensitivity compared with control cells suggesting a defect in DNA repair. Overall, NLRP3 deficient cells are more resistant to etoposide-induced apoptosis under conditions of acute genotoxic stress due to impaired p53 pathway activation, and display some sensitivity in a short-term DNA repair test. This latter phenotype is currently under investigation in different cellular models that are more appropriate for testing DNA repair

HBEC3-KT were transfected either with siCTL or siNLRP3 then exposed to 1 μ M Eto for 3 hours. Medium was changed and cells were grown for 11 days before crystal violet staining

capacity.

- The authors start with the analysis of NLRP3 in cancer prior to studying the ATM connection, which correspond to the central topic of the manuscript. However, the link between these two parts is not really clearly explained. The authors might want to consider starting with the ATM connection analyses, then to switch the studies in cancer at the end (they did this organization in the discussion).

The fact that *NLRP3* was reported as frequently altered in genome-wide studies in cancer was the starting point of our study. Our findings that its expression is mostly down-regulated in cancer led us to explore for novel functions. Since DDR gene expression is frequently down-regulated in cancers and since ATM was functionally linked to NLRP3 activation in inflammatory context, we therefore investigated if NLRP3 could be an actor in the DDR and required for ATM activation.

Concerning the text organisation since it was not raised by reviewer 2 and for the reasons detailed above, we have kept the original order.

There are also several points to consider: - what is the part of inflammasome defect and of DDR defect in the oncogenesis? For instance, what are the expressions of other inflammasome genes (ASC, caspase1) in the same cohorts? Are they co-deregulated with NLRP3, or in contrast, they are deregulated in different tumours? What is the

consequence of the other inflammasome genes in cancer? This also should be discussed.

In the figure 1A the immunoblot showed that expression of caspase-1 and ASC was not as frequently down-regulated as NLRP3 expression in NSCLC. As requested by the reviewer, we performed different analyses in several cancer cohorts using the public TCGA database. Concerning *ASC/PYCARD* it is down-regulated in LUAD but up-regulated in Breast cancer (BC) while *Caspase-1* expression is impaired in both LUAD and BC. Thus, both gene expressions are also altered in cancers, suggesting that they may be involved in cancer development. They may be involved in different pathways for instance caspase-1 or ASC down-regulation may allow cells to escape pyroptosis, also ASC loss was previously shown to promote cell proliferation (PMID: 23090995). As the subject of our manuscript is the demonstration that the novel function of NLRP3 in DDR is inflammasome independent, we have not added these data to the manuscript. However as requested also by our second reviewer we added in suppl Fig.1F the analysis of NLRP3 expression in other cancers: Pan-cancer, colon, pancreas and prostate for which NLRP3 expression is also significantly down-regulated.

Expression analysis in TCGA database of LUAD and breast cancers of Caspase-1 and ASC expressions.

- lines 211-215: The fact that that inhibition of both NLRP3 and ATM is more sensitive than single inactivation suggest that they do not act in an epistatic way? This would be in contradiction with the rest of the manuscript.

The statistical comparison on our graph was not correct, we have corrected it. Therefore, there is no difference between siCTL+ATMi and siNLRP3+ATMi, thus they do act in an epistatic manner. We thank the reviewer for noticing the mistake. We have corrected the figure and text accordingly.

- line280. (Hu et al, èà&5); I think this is not the correct citation, or Flavall's citation is missing.

We thank the reviewer for noticing the error in the citation, this has been corrected.

• - Figure 1C. There is a problem with the loading control; there is more actin signal that

can account for the fact that there is also more NLRP3 in HMEC-tert cell than in the other cell lines.

We have added the ratio NLRP3/Actin below the immunoblot which highlights that in the rare cell lines analyzed that maintained NLRP3 expression, still the amount was lower than in HMEC-hTERT.

Reviewer #2 (Comments to the Authors (Required)):

Understanding the ATM regulation is of crucial interest, with direct implication for cancer research/therapy. Presented manuscript aspires to bring another layer of complexity to ATM regulation: suggests functional link between native immunity component NLRP3 (constituent of inflammasome) and activation of ATM in response to DNA damage.

Using two cancer models, namely broncho-epithelial cells and breast cancer cells, authors showed decreased ATM activation in absence of NLRP3 after etoposide or ionizing radiation DNA damage-inducing treatment. Analysis of downstream effectors of ATM (including phosphorylated H2AX, Chk2, p53) confirmed the NLRP3 requirement. Furthermore, in absence of NLRP3 significantly lower number of gH2AX and P-ATM foci was detected, in accordance with ATM activation impairment. Evidence provided here is convincing, further corroborated using regulatable expression of NLRP3 in non-small cell lung cancer and breast cancer cell lines.

Next, the authors addressed whether NLRP3-coupled inflammasome activity is involved in response to DNA damage/repair. Based on the data, it seems that NLRP3 involvement in ATM activation is independent of the functional inflammasome activity and that DNA damage induced does not activate inflammasome effectors. Conclusion points to the new, inflammasome-independent role of NLRP3.

Regarding the attenuated ATM activity resulting in decreased P-p53 in absence of NLRP3, the apoptosis inhibition was investigated, and confirmed. However, besides cell death, other downstream effects of ATM signalling impairment could have been addressed. Taking into account decrease in DNA damage signaling and lowered number of DNA repair foci resulting from NLRP3 absence, overall impairment of DNA repair kinetics (and effectivity) could be expected. Addressing the DNA repair characteristics of induced DNA damage in the absence of NLRP3 may provide direct evidence of functional consequences of NLRP3 downregulation. Long-term effects of DNA repair impairment should be reflected also on the genome in/stability of NLRP3-compromised cells.

An effort to provide mechanistic clue on NLRP3-ATM interaction is the subject of the final part of the experimental work. The pull-down data showing the ATM-NLRP3 interaction seem to be trustworthy. Less convincing I found the results of Proximity Ligation Assay, as majority of foci (see Fig.6H, not treated sample on the left) are located outside the nuclei. Conclusion based on these data Moreover, the role of cytoplasmic ATM is considered to be different from nuclear ATM function.

Altogether, manuscript brings significant claims on interesting topic and presents valuable contribution to the field. To improve the quality of the manuscript I suggest following points to be addressed:

We thank the reviewer for appreciating the novelty of our findings.

1. To strengthen the conclusion of the paper, addressing the possible DNA repair impairment would be beneficial. One option is to bring direct evidence of changes in DNA

damage removal/persistence in NLRP3-compromised cells (comet assay may be the tool), another way is to provide the data supporting the evidence of increased genomic instability as a consequence of long-term ATM activity impairment caused by NLRP3 down-regulation.

The role of NLRP3 in genome stability is an aspect we are currently studying and will be the subject of a future publication. However as requested, we added the analysis on public databases that correlates the level of NLRP3 expression with genome instability in LUAD and in breast cancer. These data have been added in Suppl Figure 2A and highlight that the expression of NLRP3 is positively associated with genome stability, which strengthen the message that NLRP3 is a protein involved in the DDR.

2. Regarding the quantification of foci in Proximity Ligation Assay (Fig.6H), the question is whether the foci were counted only in DAPI-stained nuclear area, or in whole cell including cytoplasm. The immunofluorescence image displaying only the red channel for the same area should be included in the final Fig.6H to provide the evidence of nuclear foci, preferably in higher magnification. Currently presented merged images in Fig.6H do not allow for verification of nuclear foci presence.

We counted the PLA signals of the whole cells since the signal is detected in both compartments, and also decreased in both upon DSB formation. As discussed for reviewer 1 and based on our preliminary data suggesting that NLRP3 and ATM shuttles together to the nucleus in a rhythmic manner (data not shown), we speculate that the shuttling of NLRP3 to the nucleus is tightly controlled and involved in the activation of ATM by DNA double strand breaks. This is currently under investigation. As requested we have added on the panel the red channel only of the same area displaying PLA staining.

3. The experimental part of the manuscript is focused on 2 types of cancer. To help appreciate the scale of NLRP3 dysregulation in cancer pathogenesis, the information on another types of cancer associated with NLRP3 could be included in the paper.

This point was also raised by reviewer 1 therefore we performed a more global analysis of NLRP3 expressions in different cancer cohorts using public database. These results are shown Suppl fig 1F. Beside LUAD and breast cancers, NLRP3 is also down-regulated in pan-cancer, colon, prostate and pancreatic cancers.

January 9, 2023

RE: Life Science Alliance Manuscript #LSA-2022-01494-TR

Dr. Virginie Petrilli
Cancer Center of Lyon
28 rue Laennec
Lyon 69008
France

Dear Dr. Petrilli,

Thank you for submitting your revised manuscript entitled "Inflammasome-independent NLRP3 function enforces ATM activity in response to genotoxic stress.". We would be happy to publish your paper in Life Science Alliance pending final revisions necessary to meet our formatting guidelines.

-please add an ethics statement for your human patient samples

A. FINAL FILES:

B. MANUSCRIPT ORGANIZATION AND FORMATTING:

****It is Life Science Alliance policy that if requested, original data images must be made available to the editors. Failure to provide**

original images upon request will result in unavoidable delays in publication. Please ensure that you have access to all original data images prior to final submission.**

The license to publish form must be signed before your manuscript can be sent to production. A link to the electronic license to publish form will be sent to the corresponding author only. Please take a moment to check your funder requirements.

Sincerely,

Reviewer #1 (Comments to the Authors (Required)):

The authors have addressed satisfactorily my comments.

Reviewer #2 (Comments to the Authors (Required)):

I appreciate author's effort and time invested in improving the manuscript.

Regarding the points raised by Reviewer #2, each was correctly addressed and approached by the authors.

In case of point #1, authors decided to include new Figures (Suppl Fig 2A and Suppl Fig 2B), thus providing evidence that decreased NLRP3 expression positively correlated with genome instability in both LUAD and BC.

In point #2, the improvements in data presentation of Proximity Ligation Assay (Fig.6H) were demanded. As a result, now the Fig.6H contains high-quality fluorescent-microscopy images and authors incorporated the red-channel-only on the panel Fig.6H. In addition, as part of response to Reviewer#1, data-based hypothesis on the role of NPL3 in the complex with ATM was extended in Discussion.

Point #3 led the authors to include databases-derived data showing the NLRP3 down-regulated expression also in pan-cancer, colon, prostate and pancreatic cancers (Suppl Fig 1F).

Altogether, points were satisfactory answered and particular responses contributed to improved manuscript quality.

January 20, 2023

RE: Life Science Alliance Manuscript #LSA-2022-01494-TRR

Dr. Virginie Petrilli
Cancer Center of Lyon
28 rue Laennec
Lyon 69008
France

Dear Dr. Petrilli,

Thank you for submitting your Research Article entitled "Inflammasome-independent NLRP3 function enforces ATM activity in response to genotoxic stress.". It is a pleasure to let you know that your manuscript is now accepted for publication in Life Science Alliance. Congratulations on this interesting work.

DISTRIBUTION OF MATERIALS:

Again, congratulations on a very nice paper. I hope you found the review process to be constructive and are pleased with how the manuscript was handled editorially. We look forward to future exciting submissions from your lab.

Sincerely,
